# Increasing Mean Age of Head and Neck Cancer Patients at a German Tertiary Referral Center

**DOI:** 10.3390/cancers13040832

**Published:** 2021-02-17

**Authors:** Julius M. Vahl, Marlene C. Wigand, Michael Denkinger, Dhayana Dallmeier, Chiara Steiger, Claudia Welke, Peter Kuhn, Christian Idel, Johannes Doescher, Adrian von Witzleben, Matthias Brand, Ralf Marienfeld, Peter Möller, Marie-Nicole Theodoraki, Jens Greve, Patrick J. Schuler, Cornelia Brunner, Thomas K. Hoffmann, Simon Laban

**Affiliations:** 1Department of Otorhinolaryngology and Head & Neck Surgery, Head and Neck Cancer Center of the Comprehensive Cancer Center Ulm, University Medical Center Ulm, 89075 Ulm, Germany; julius.vahl@uniklinik-ulm.de (J.M.V.); marlene.wigand@uniklinik-ulm.de (M.C.W.); chiara.steiger@uni-ulm.de (C.S.); johannes.doescher@uniklinik-ulm.de (J.D.); adrian.vonwitzleben@uniklinik-ulm.de (A.v.W.); matthias.brand@uniklinik-ulm.de (M.B.); marie-nicole.theodoraki@uniklinik-ulm.de (M.-N.T.); jens.greve@uniklinik-ulm.de (J.G.); patrick.schuler@uniklinik-ulm.de (P.J.S.); Cornelia.brunner@uniklinik-ulm.de (C.B.); t.hoffmann@uniklinik-ulm.de (T.K.H.); 2Agaplesion Bethesda Ulm, Geriatric Research Ulm University and Geriatric Center, 89073 Ulm, Germany; michael.denkinger@bethesda-ulm.de (M.D.); dhayana.dallmeier@bethesda-ulm.de (D.D.); 3Clinical Cancer Registry, Comprehensive Cancer Center Ulm, University Medical Center Ulm, 89081 Ulm, Germany; claudia.welke@uniklinik-ulm.de (C.W.); peter.kuhn@uniklinik-ulm.de (P.K.); 4Department of Otorhinolaryngology and Head & Neck Surgery, University Medical Center Schleswig-Holstein, Campus Lübeck, 23562 Lübeck, Germany; Christian.Idel@uksh.de; 5Cancer Sciences Unit, Faculty of Medicine, University of Southampton, Southampton SO17 1BJ, UK; 6Institute of Pathology, University Medical Center Ulm, 89081 Ulm, Germany; ralf.marienfeld@uniklinik-ulm.de (R.M.); peter.moeller@uniklinik-ulm.de (P.M.)

**Keywords:** aging, head and neck cancer, demographics, epidemiology, HPV

## Abstract

**Simple Summary:**

We detected an increase in the mean age at diagnosis among 2450 patients with head and neck carcinoma diagnosed between 2004 and 2018 in the head and neck cancer center Ulm (Germany) in comparison to the mean age in the general population of the main catchment area. Carcinoma incidences were rising too. The steepest ascent in incidence rates was found in the age group >70 years of age. These results indicate that health care providers need to prepare for an increasingly older group of cancer patients. This study is the first step to a nationwide analysis among German patients.

**Abstract:**

Background: The impact of demographic change on the age at diagnosis in German head and neck cancer (HNC) patients is unclear. Here we present an evaluation of aging trends in HNC at a tertiary referral center. Methods: Retrospective cohort study on aging trends at the initial diagnosis of newly diagnosed patients with HNC between 2004 and 2018 at the head and neck cancer center Ulm in relation to demographic data of the catchment area. Results: The study population consisted of 2450 individuals diagnosed with HNC with a mean age of 62.84 (±11.67) years. We observed a significant increase in annual incidence rates and mean age over time. Mean age among HNC patients increased significantly more than among the population in the catchment area. Whereas the incidence rate of patients <50 years did not change, the incidence of HNC patients aged ≥70 years increased the most. The mean patient age in the main tumor sites increased significantly. Surprisingly, HPV-positive patients were not younger than HPV-negative patients, but showed a non-significant trend towards a higher mean age (63.0 vs. 60.7 years). Conclusions: Increasing incidence rates in older patients pose a challenge for health care systems. A nationwide study is needed to assess the dynamics and impact of aging on the incidence of HNC.

## 1. Introduction

Head and neck cancer (HNC) represents the sixth most common malignancy worldwide [1]. The major risk factors are nicotine abuse, excessive alcohol consumption and human papillomavirus infection [2]. In addition, older age is a predisposing factor gaining increasing importance [3]. According to data from the “World Population Prospects” in 2018, persons aged 65 years or more outnumbered children under the age of five for the first time in history [4]. They make up the fastest-growing age group and until 2050, their number is projected to more than double. Moreover, the percentage of all cancers diagnosed in older adults is estimated to rise significantly [5]. Accordingly, the number of older patients diagnosed with HNC is likely to increase in the future as well. Incidences are expected to rise in Central and Eastern Europe, Scandinavia, Japan, Australia and Canada [2]. Hypertension, hyperlipidemia, chronic obstructive pulmonary disease and diabetes are among the most common comorbidities in older HNC patients [6]. They go along with decreased nutritional and performance status and impaired organ function and thus limit treatment options. Moreover, the immune system experiences numerous changes with increasing age that affect both the innate and the adaptive immune system. These changes are generally referred to as immunosenescence and contribute to a decreased immunosurveillance in older patients [7,8]. On the other hand, this demographic development is countered by an increasing incidence of human papillomavirus (HPV) associated oropharyngeal squamous cell carcinomas (OPSCC) [9,10,11]. HPV-associated OPSCC patients are considered to be younger than HPV-negative patients [12]. Nevertheless, an increasing fraction of older patients are diagnosed with an HPV-associated OPSCC [13]. The aim of the present study was to assess developments in the age at diagnosis among HNC patients at a large tertiary referral center within a timeframe of 15 years.

## 2. Materials and Methods

### 2.1. Patient and Demographic Data

Patient data were extracted from the clinical cancer registry of the comprehensive cancer center Ulm (CCCU) for all patients with an initial diagnosis of HNC within the period from 01 January 2004 until 31 December 2018. The cancer registry of the CCCU has been established in the year 2000. Since then, all newly diagnosed cases have been registered into the database. The registration of all cancer cases into cancer registry databases has become legally binding since 2009. Epidemiologic data reports for research purposes can be accessed. Patients are routinely informed about their right of objection against the collection of personal data. Parameters extracted were: age at diagnosis in years, gender, primary tumor site, TNM-classification and human papillomavirus status. Data were analyzed in an anonymized format. Patients diagnosed with squamous cell carcinoma of the skin and tumor types others than mucosal SCC in the head and neck region were not included. Postal codes of patients were analyzed and graphed to define the total catchment area (Figure 1A). The main catchment area was defined based on the boarders of 27 counties as shown in Figure 1B. Anonymized demographic data of the population within the main catchment area was obtained from the federal statistical office (Statistisches Landesamt) of Baden-Württemberg and Bavaria. The mean age in the general population of these 27 counties was averaged and the standard deviation of the mean age in the main catchment area was calculated.

Absolute numbers were cumulative counts of patients diagnosed per annum. Incidence rates per 100,000 were calculated based on the number of newly diagnosed cased in relation to the respective population at risk utilizing the demographic data of the main catchment area. Relative numbers were calculated using the absolute number of the respective subgroup as the numerator and the total number of annually diagnosed patients as the denominator.

HPV-status was determined by multiplex HPV-DNA PCR (GP5+/GP6+ primers followed by Sanger sequencing for HPV typing) as previously described [14] and p16 immunohistochemistry was performed. Patients were considered HPV-positive if both tests were positive.

### 2.2. Statistical Analysis

All statistical analyses were performed using SPSS (Version 26) and GraphPad Prism (Version 8.4.3.). Data was tested for Gaussian distribution using the Shapiro–Wilk test. A one-way ANOVA was used to test for mean differences between more than two groups. The mean age in every following year starting in 2005 was compared to the mean age observed in 2004 using Dunnett’s multiple comparison test. Patients were grouped by T-status, N-status, gender and into age groups (group 1: <50 years, group 2: ≥50 <70 years and group 3: ≥70 years). Incidence rates were calculated based on demographic data within the respective group in the main catchment area. Linear regression analysis was performed to analyze and compare incidence rates per 100,000 inhabitants. The mean age by the primary tumor site was compared to the mean age of any other primary tumor site using the Tukey’s multiple comparison test. Differences between two groups with normal distribution were performed using the Student’s *t*-test. *p*-values were corrected using the two-stage linear step-up procedure of Benjamini, Krieger and Yekutieli with a false discovery rate of 5% [15].

## 3. Results

In total, 2450 patients were diagnosed with HNC between 2004 and 2018 and included in this analysis (on average 163.3 patients per year). The mean age was 62.8 (standard deviation: ±11.7) years. Detailed patient characteristics subgrouped in 5-year intervals are presented in Table 1. Age at diagnosis was significantly different by primary site and N-status. Nasopharyngeal patients were the youngest and middle ear carcinoma patients were the oldest. Patients with positive nodal status were significantly younger than N0 patients. Within the main catchment area, the population grew from 5.08 million inhabitants in 2004 to 5.23 million inhabitants.

The mean age per year of diagnosis differed significantly over the evaluated time period (F = 3.483, *p* < 0.0001). Compared to the mean age in 2004 (59.4 years) a significantly higher mean age at diagnosis was determined in 2014 (64.3 years), 2015 (64.8 years), 2016 (63.7 years), 2017 (65.1) until 2018 (63.9 years). The absolute difference between the mean age in 2004 and 2018 was 4.5 years. At the same time, the incidence rates per 100,000 inhabitants in the catchment area increased from 2.40 in 2004 up to 4.84 in 2018. This increase was significant (R^2^ = 0.91, slope: 0.19 (95% CI: 0.15–0.22), *p* < 0.0001). The age distribution and incidence rates of newly diagnosed cases per 100,000 inhabitants per year are visualized in Figure 2A. During the same period, the mean age in the catchment area increased from 41.0 to 43.8 years (delta: 2.8 years; R^2^ = 0.94, slope: 0.21 (95% CI: 0.18–0.24), *p* < 0.0001). Compared to the increase in mean age among HNC patients (R^2^ = 0.87, slope: 0.37 (95% CI: 0.27–0.45), *p* < 0.0001) the slopes were significantly different (*p* = 0.0008; Figure 2B).

Localized (cT1/cT2) and locally advanced tumors (cT3/cT4) both showed a rising incidence rate (cT1/cT2: R^2^ = 0.87, slope: 0.12 (95% CI: 0.09–0.15), *p* < 0.0001; cT3/cT4: R^2^ = 0.85, slope: 0.11 (95% CI: 0.08–0.14), *p* < 0.0001) (Figure 3A). The slopes were not significantly different between cT1/cT2 and cT3/cT4. The mean ratio of T1/2 to T3/4 was 0.87 (median = 0.87, standard deviation = 0.147) indicating that locally advanced tumors had a higher incidence throughout the whole period.

Both, incidence rates of N0 (R^2^ = 0.74, slope: 0.10 (95% CI: 0.07–0.14); *p* < 0.0001) and N+ patients (R^2^ = 0.90, slope: 0.12 (95% CI: 0.09–0.15); *p* < 0.0001) increased significantly, whereas the slopes were not significantly different (Figure 3B). The ratio of N0/N+ patients was 0.72 (median = 0.75, standard deviation = 0.17) indicating that the majority of patients were diagnosed with a positive nodal status.

Regarding gender distribution in our cohort we saw a clear and consistent predominance of male patients in regard to incidence rates (Figure 3C). However, incidence rates in males (R^2^ = 0.80, slope: 0.26 (95% CI: 0.18–0.34); *p* < 0.0001) and females increased significantly (R^2^ = 0.77, slope: 0.10 (95% CI: 0.07–0.14); *p* < 0.0001), whereas the slope was significantly higher in male patients (*p* = 0.0004). The mean ratio of male to female patients was 4.24 (median = 3.81, standard deviation = 1.58, range = 2.73–8.38).

Both in age group 2 (≥ 50 < 70 years: (R^2^ = 0.80, slope: 0.31 (95% CI: 0.22–0.40), *p* < 0.0001) and group 3 (>70 years: R^2^ = 0.87, slope: 0.51 (95% CI: 0.39–0.63), *p* < 0.0001), a significantly rising incidence rate was detected, whereas the incidence rate of younger patients (<50 years; group 1) did not change significantly (Figure 3D). The slope of incidence rates between group 2 and 3 was significantly different (*p* = 0.0078).

The mean age differed significantly between primary tumors sites (ANOVA: *p* < 0.0001, Table 1). The age distribution by the primary site is shown in Figure 4A. Nasopharynx carcinoma patients formed the youngest subgroup < 60 years of age (median = 55.3 years). Patients with oropharynx (median = 60.8 years), sinunasal (median = 60.9 years) oral cavity (median = 61.6 years) and hypopharynx cancer (median = 61.9 years) formed a middle group between 61 and 62 years. Patients with laryngeal (median = 66.0 years), salivary gland (median = 70.9 years) and lip cancer (median = 75.0 years) formed the oldest patient group ≥ 65 years. In summary, significant age differences were primarily seen between the nasopharynx subgroup and the primary sites of the other two age groups- however, also, between the middle age group and the oldest primary sites.

Within the most prevalent four primary tumors sites, the mean age increased significantly for oropharynx (R^2^ = 0.85, slope: 0.37 (95% CI: 0.28–0.46), *p* < 0.0001) and hypopharynx (R^2^ = 0.71, slope: 0.59 (95% CI: 0.37–0.82), *p* < 0.0001), larynx (R^2^ = 0.32, slope: 0.29 (95% CI: 0.04–0.54), *p* = 0.0267, *q* = 0.028) and oral cavity (R^2^ = 0.37, slope: 0.37 (95% CI: 0.08–0.66) *p* = 0.0176, *q* = 0.025). These results are graphed in Figure 4B.

Among all 760 oropharyngeal cancers, HPV-status results were available for 328 patients. Of these 196 were HPV-negative and 132 were HPV-positive. Mean age at diagnosis in HPV-negative patients was 60.7 years (range 38.2–78.2 years) and 63.0 years (range: 35.7–90.7 years) in HPV-positive patients, this was not a statistically significant difference (*p* = 0.025, *q* = 0.105). The age distribution by HPV-status is graphed in Figure 5A. The mean age by HPV-status was analyzed. There was no significant increase in the mean age (HPV-negative: R^2^ = 0.0029, *p* = 0.496; HPV-positive: R^2^ = 0.0057, *p* = 0.407). The mean age by HPV-status with standard deviation is graphed in Figure 5B.

## 4. Discussion

We demonstrated a significant increase in the mean age at the diagnosis of HNC over a period of 15 years. To the best of our knowledge, this is the first study focusing on the demographic development of HNC patients in Germany. The absolute difference between the mean age in 2004 and 2018 was 4.5 years. In comparison, the mean age in the main catchment area increased by only 2.8 years. The slopes of the two linear regression models were significantly different. This indicates that the mean age among patients with HNC is increasing more rapidly than in the general population.

Similar results have recently been reported from the USA with a slightly different methodology and timeframe [16]. The group reported steadily increasing age among non-oropharyngeal cancers, whereas oropharyngeal cancers demonstrated a fluctuating age development. Age data and distribution seem very similar to the data found in our study suggesting that these developments are in fact disease related. The group did not have access to HPV-status data from the Surveillance, Epidemiology, and End Results (SEER) database. However, the fraction of HPV-positive patients in the USA is even higher than in Germany. 

Our data reveal continuously growing numbers of HNC patients at a tertiary referral center for head and neck cancer in general and of older HNC patients in particular. The increasing numbers of HNC patients were noted for early and advanced tumors, both sexes and all primary tumor sites. By far, the steepest rise was observed in the incidence rate among patients >70 years of age.

Centralization of public health care is performed worldwide and increasing numbers of patients are treated in specialized head and neck cancer centers in Germany. At the same time, the incidence of oropharyngeal and oral cavity cancer has been shown to rise in numerous countries worldwide [17]. Authors attributed the increasing numbers of oropharyngeal cancer to the emerging role of human papillomavirus (HPV) as a carcinogenic risk factor [18]. The recommendation of prophylactic HPV vaccination in both boys and girls has been established in Germany since 2018, but vaccination rates with the three recommended doses in adolescent females where the vaccine has been available since 2007 are below 40% [19]. It is expected that OPSCC numbers will continue to rise for another 15–20 years before effects of the vaccine programs may become evident. Having in mind a presumably younger age at diagnosis in virally induced OPSCC [13], decreasing age in the oropharyngeal cancer patients could be expected. Interestingly, nasopharyngeal and oropharyngeal cancer in fact have the lowest mean age in this cohort. However, the mean age in the OPSCC cohort is also increasing significantly over time. Unfortunately, HPV-status was not available for all oropharyngeal patients. HPV-testing has been introduced at our site in 2013. However, only since 2017 it has become a standard for routine diagnostics of OPSCC according to the international guidelines (AJCC Cancer Staging Manual version 8, 2017). Thus, only in a subgroup of OPSCC patients diagnosed between 2012 and 2018, HPV-status was available. Among these, no significant differences regarding age at diagnosis were observed. On the other hand, in the total cohort of OPSCC patients there was also no significant trend towards increasing mean age if the analysis was limited to the period from 2012 to 2018. Results from the SEER database [16] and from other studies [13,20] also challenge the paradigm of the generally younger HPV-positive patient.

However, we saw a non-significant higher maximum age and higher mean age in HPV-positive patients. Data of other groups have also indicated increasing HPV-positivity among older OPSCC patients [13]. Within the timeframe covered in this analysis, there was neither a clear trend towards increasing mean age among HPV-negative or HPV-positive patients. Further studies in even larger cohorts are needed to elucidate this question.

Some authors stated in their projection model study that the incidence of most cancers in Germany is projected to increase significantly until 2030 [21]. The group of HNC patients aged 70 years and older grew strongest in our study population. Older patients tend to have a higher rate of comorbidity (55–98%) associated with a poor quality of life and high health care costs [22]. Consequently, the tolerability of intensive treatment is much lower in elder patients. Thus, treatment-related deaths are higher in older adults, while the short-term outcome seems not to be affected [23].

Ulm is a medium large city with approximately 127,000 inhabitants, but is localized at the center of a large rural area with an 80–150 km patient outreach resulting in a large reference population in the catchment area. The majority of patients diagnosed at our center travel >30 km. One source of bias is the potentially incomplete registration of cases in the main catchment area for two reasons. First of all, in Germany patients can be diagnosed and treated not only in cancer centers, but also in any hospital. Thus, it is likely that the number of cases registered at the CCCU cancer registry does not cover the complete incidence within the catchment area. The spiking incidence rate may be partly due to an increasing referral of cancer patients to large comprehensive cancer centers. Secondly, the total catchment area did not follow county boarders. Thus, the definition of the main catchment area is a potential bias, because the reference population may have been overestimated. On the other hand, demographic data are only available for whole counties. In consequence, in a future study, nationwide patient cohorts should be compared with respect to small, medium, and large cities and metropolitan areas in different geographic regions of Germany to map a conclusive picture of HNC patient demographics in Germany.

## 5. Conclusions

The increasing annual incidence rates and mean age of HNC patients at a tertiary referral center within 15 years reveal the need for more specific focus on elderly HNC tumor patients. We conclude that the management of increasingly older patients will require specialized cancer centers tailoring treatments to the needs of older cancer patients. A national analysis may be needed to assess the impact of urban compared to rural demographics with regard to the dynamics and impact of patient age at diagnosis in HNC patients.

## Figures and Tables

**Figure 1 cancers-13-00832-f001:**
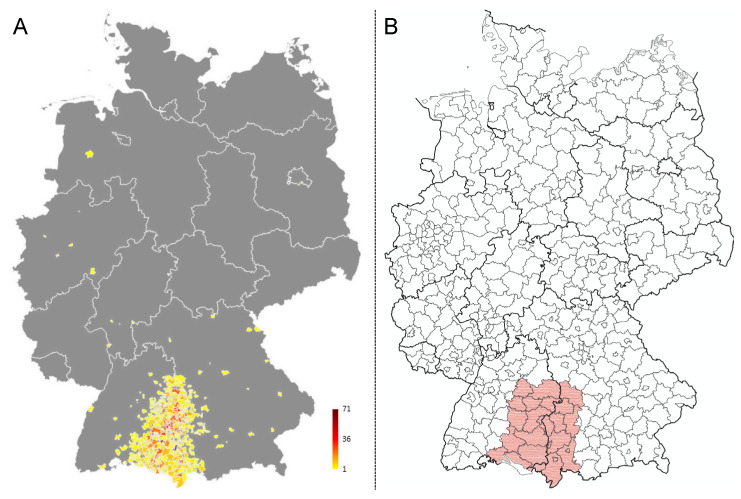
Total and main catchment area of the patient cohort. (**A**) Patient numbers were graphed to the map of Germany based on their postal codes using Excel. The colors indicate the number of patients per dot as indicated in the figure legend. (**B**) The main catchment area was defined based on county borders to be able to obtain demographic population data of the main catchment area (indicated in red).

**Figure 2 cancers-13-00832-f002:**
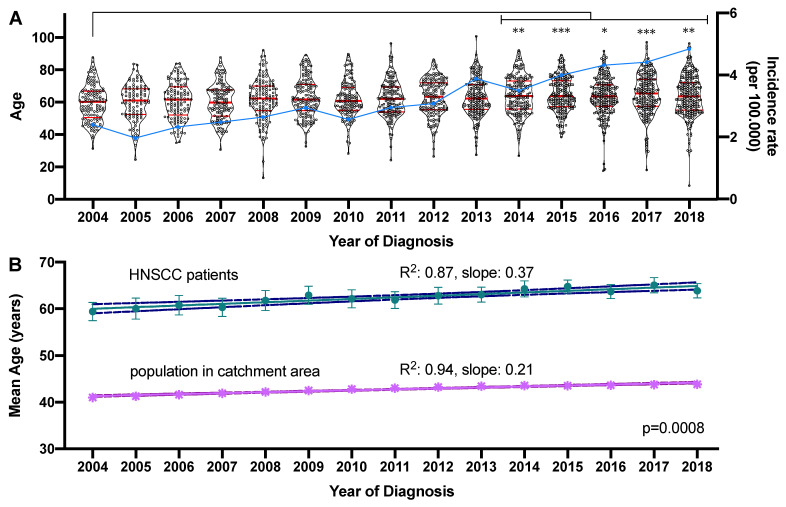
Development of the age at diagnosis and newly diagnosed cases between 2004 and 2018. (**A**) The mean age of the 2450 patients diagnosed between 2004 and 2018 is graphed on the left y-axis as a violin plot showing all individual values. The median and quartiles are indicated by red horizontal bars. Significant changes in the mean age compared to 2004 are indicated by asterisks. The incidence rate of newly diagnosed cases per 100,000 is graphed on the right y-axis in the form of a blue line. * adjusted *p*-value < 0.05, ** adjusted *p*-value < 0.01, *** adjusted *p*-value < 0.001. (**B**) Linear regression model of the mean age in the HNC patient cohort and the population in the main catchment area. The mean age ± standard deviation is indicated by the symbols and the linear regression model is shown as the connecting line with the upper and lower 95% confidence interval indicated by broken lines.

**Figure 3 cancers-13-00832-f003:**
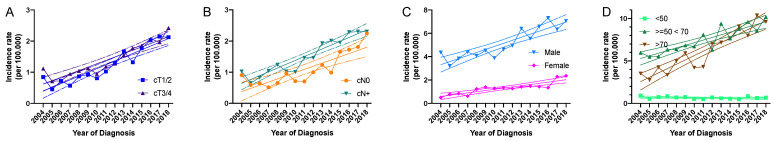
Absolute and relative numbers of HNC patients between 2004 and 2018. All data are plotted showing the mean. (**A**–**D**): Incidence rates of HNC patients per 100,000 in the respective population at risk are graphed on the *y*-axis and the year of diagnosis on the *x*-axis by (**A**) T-status (T1/T2 vs. T3/T4), (**B**) nodal status (N0 vs. N+), (**C**) gender and (**D**) age (<50, ≥50 <70 and ≥70 years). The lines indicate the linear regression model and the broken lines the upper and lower 95% confidence interval.

**Figure 4 cancers-13-00832-f004:**
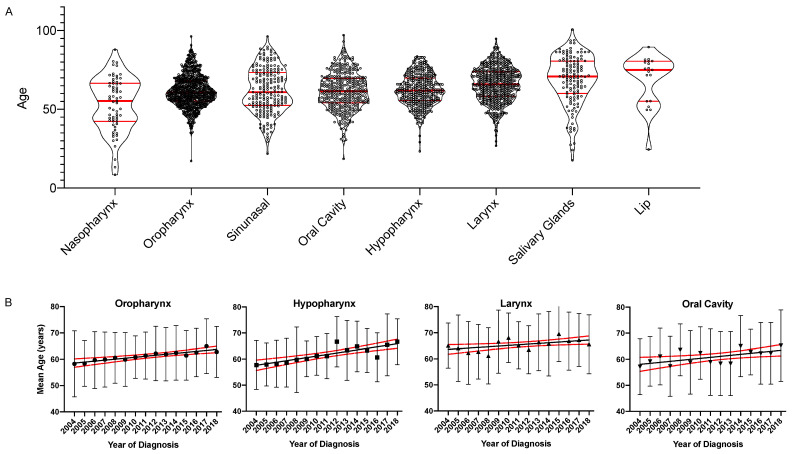
Grouped scatter plots and linear regression of age and year of diagnosis separated by the primary tumor site. (**A**). The mean age of patients diagnosed from 2004 to 2018 by primary tumor site is graphed on the left *y*-axis as a violin plot showing all individual values. Patient subgroups were ordered by the median age. Median and quartiles are indicated by red horizontal bars. (**B**). The mean age ± standard deviation for the most prevalent four primary tumor sites (oropharynx, hypopharynx, larynx and oral cavity) is graphed on the *y*-axis over time on the *x*-axis. The lines indicate the linear regression model and the broken lines the upper and lower 95% confidence interval.

**Figure 5 cancers-13-00832-f005:**
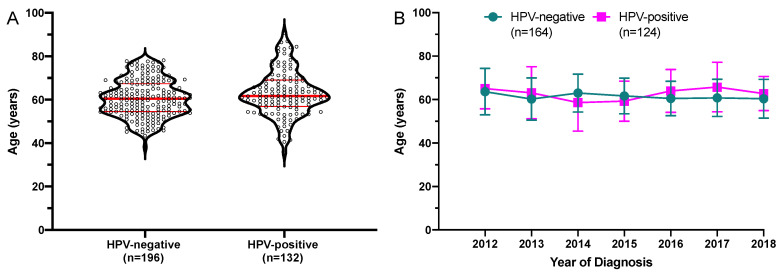
Age at diagnosis for the subgroup of oropharyngeal cancers with known HPV-status. (**A**) The mean age of patients diagnosed from 2004 to 2018 HPV-status is graphed on the left y-axis as a violin plot showing all individual values. The median and quartiles are indicated by red horizontal bars. Mean age at diagnosis was not significantly different. HPV-negative: *n* = 196, HPV-positive: *n* = 132. (**B**) Mean age with standard deviation of a subgroup of patients diagnosed between 2012 and 2018 for who the HPV-status was available. Only diagnosis years with ≥5 patients in both subgroups were included in this analysis.

**Table 1 cancers-13-00832-t001:** Patient characteristics.

Time Period		2004–2008			2009–2013			2014–2018			Total: 2004–2018			*p*-Value
Variable		*N*	%	Mean Age	*N*	%	Mean Age	*N*	%	Mean Age	*N*	%	Mean Age	
**Gender**	Male	476	83.2	60.5	582	77.8	63.1	806	77.8	64.6	1944	79.3%	62.9	0.781
	Female	96	16.8	62.3	166	22.2	61.9	230	22.2	63.6	506	20.7%	62.7	
	Total	572	100.0	60.8	748	100.0	62.8	1036	100.0	64.4	2450	100.0%	62.8	n.a.
**Primary site**	Oropharynx	166	29.0	59.7	231	30.9	61.3	342	33.0	62.8	760	31.0%	61.5	<0.0005
	Hypopharynx	107	18.7	58.5	103	13.8	62.8	140	13.5	64.5	372	15.2%	62.1	
	Larynx	147	25.7	63.7	183	24.5	65.8	230	22.2	67.1	570	23.3%	65.7	
	Oral Cavity	66	11.5	60.8	111	14.8	60.1	135	13.0	64.3	350	14.3%	61.7	
	Sinunasal	39	6.8	61.3	56	7.5	63.8	96	9.3	61.4	193	7.9%	62.0	
	Nasopharynx	16	2.8	55.6	23	3.1	54.9	20	1.9	49.7	59	2.4%	53.3	
	Salivary Glands	27	4.7	65.3	37	4.9	67.5	59	5.7	71.4	124	5.1%	68.8	
	Lip	4	0.7	51.4	4	0.5	75.6	10	1.0	71.5	18	0.7%	67.9	
	Middle Ear	0	0.0		0	0		4	0.4	71.1	4	0.2%	71.1	
**T-status**	cT1/2	162	28.3	62.4	274	36.6	63.2	455	43.9	63.1	944	38.5%	62.7	0.087
	cT3/4	228	39.9	60.2	300	40.1	62.6	504	48.6	65.7	1052	42.9%	63.5	
	missing	182	31.8		574	76.7		77	7.4		454	18.5%		n.a.
**N-status**	cN0	160	28.0	64.1	217	29	63.2	415	40.1	66.0	830	33.9%	65.3	<0.0005
	cN+	235	41.1	59.0	342	45.7	62.6	538	51.9	63.0	1143	46.7%	61.3	
	missing	177	30.9		189	25.3		83	8.0		449	18.3%		
**Age grouped**	<50	109	19.1	43.9	86	11.5	43.9	94	9.1	42.2	307	12.5%	43.5	n.a.
	≥50 < 70	342	59.8	60.7	452	60.4	59.9	599	57.8	60.6	1461	59.6%	60.4	
	>70	121	21.2	76.4	210	28.1	76.7	343	33.1	77.2	682	27.8%	76.9	

## Data Availability

Anonymized datasets generated and analyzed during the current study are available from the corresponding author upon reasonable request.

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
