# Peer review of "Increasing Mean Age of Head and Neck Cancer Patients at a German Tertiary Referral Center"

_cancers, 2021, doi:10.3390/cancers13040832_

Round 1

Reviewer 1 Report

The authors have made a very interesting and original article that provides valuable information on a disease of great importance such as head and neck cancer. The article is well written and neat, although there are some aspects that could be improved. It would be very interesting to be able to include additional information. The authors in the introduction indicate that the greatest risk factors are smoking, excessive alcohol consumption, and HPV infection, but they only study HPV. It may be possible to collect from medical records whether patients are or have been smokers, and whether they consume excess alcohol, since these risk factors have not been taken into account. For example, almost in all probability, older patients will have smoked or will smoke in a greater proportion than younger ones, due to the anti-tobacco policies that have been implemented in recent decades. This reduction in the consumption of tobacco and excess alcohol consumption may be some of the causes, which increases the age at which certain types of cancer are suffered, and is the reason why I think that if possible, it should be included as one more variable. It would also be interesting to include comorbidities, as the authors also indicate in the introduction, and if they can be collected from the patients' medical records. On the other hand, there is too much data in the figures that complicates the reading, and in many cases it is not relevant, and perhaps you should turn to supplementary material, such as from figure 3E to H. For the rest of the manuscript, they provide interesting information, from a very specific area, but for a period of 15 years, and with a sufficient number of patients. I think that if the data that I suggest can be collected, they will be able to complete the study.

Reviewer 2 Report

The authors analyzed a pattern of the change in mean age of head and neck cancer patients at a German hospital in Ulm. They found that the mean age of patients increased during their observation period, from 2004 to 2018. This increase might be in accord with the increase in mean age of the entire population in the catchment area, however, the extent of increase was more pronounced than that of the entire population. They also described in the summary that HPV-positive patients were younger than HPV-negative patients.

This paper is well written, and I think this paper is of interest because these result inform us of the trend in patients demography which is likely to be true of other developed countries.

However, I think one expression in summary should be corrected. Even though there was no significant difference between the mean age in HPV-positive and HPV negative patients, as shown in the results section, the authors described that the former tended to be younger than the latter. This expression should be replaced with more objective one, just describing the result of the calculation for the mean age.

Round 2

Reviewer 1 Report

The authors have made some modifications, although they have not included the data on tobacco consumption and excess alcoholic beverages of the patients, but it has improved slightly.
Congratulations to the authors for this manuscript.